# Tad pilus-mediated twitching motility is essential for DNA uptake and survival of Liberibacters

Lulu Cai[1][☯], Mukesh Jain[1][☯], Marta Sena-Vélez[2], Kathryn M. Jones[2], Laura A. Fleites[3], Michelle Heck[3], Dean W. Gabriel[1]*

1 Plant Pathology Department, University of Florida, Gainesville, Florida, United States of America,
2 Department of Biological Science, Florida State University, Tallahassee, Florida, United States of America,
3 USDA Agricultural Research Service, Robert W. Holley Center for Agriculture and Health, Ithaca, New York, United States of America

☯ These authors contributed equally to this work.
* dgabr@ufl.edu

**Data Availability Statement:** All relevant data are within the manuscript and its Supporting information files.

## Abstract

Axenically cultured *Liberibacter crescens* (Lcr) is a closely related surrogate for uncultured plant pathogenic species of the genus Liberibacter, including 'Candidatus L. asiaticus' (CLas) and 'Ca. L. solanacearum' (CLso). All Liberibacters encode a completely conserved gene repertoire for both flagella and Tad (Tight Adherence) pili and all are missing genes critical for nucleotide biosynthesis. Both flagellar swimming and Tad pilus-mediated twitching motility in Lcr were demonstrated for the first time. A role for Tad pili in the uptake of extracellular dsDNA for food in Liberibacters was suspected because both twitching and DNA uptake are impossible without repetitive pilus extension and retraction, and no genes encoding other pilus assemblages or mechanisms for DNA uptake were predicted to be even partially present in any of the 35 fully sequenced Liberibacter genomes. Insertional mutations of the Lcr Tad pilus genes *cpaA*, *cpaB*, *cpaE*, *cpaF* and *tadC* all displayed such severely reduced growth and viability that none could be complemented. A mutation affecting *cpaF* (motor ATPase) was further characterized and the strain displayed concomitant loss of twitching, viability and reduced periplasmic uptake of extracellular dsDNA. Mutations of *comEC*, encoding the inner membrane competence channel, had no effect on either motility or growth but completely abolished natural transformation in Lcr. The *comEC* mutation was restored by complementation using *comEC* from Lcr but not from CLas strain psy62 or CLso strain RS100, indicating that unlike Lcr, these pathogens were not naturally competent for transformation. This report provides the first evidence that the Liberibacter Tad pili are dynamic and essential for both motility and DNA uptake, thus extending their role beyond surface adherence.

**Funding:** This work was supported by the USDA National Institute of Food and Agriculture; Specialty Crops Research Initiative (NIFA-SCRI) grant #2016-70016-24844 to DWG and KMJ. https://nifa.usda.gov/funding-opportunity/specialty-crop-research-initiative-scri. The funders had no role in study design, data collection and analysis, decision to publish, or preparation of the manuscript.

**Competing interests:** The authors have declared that no competing interests exist.

## Introduction

'*Candidatus* Liberibacter' spp. are fastidious and uncultured $\alpha$-Proteobacteria (order *Rhizobiales*) that have emerged as a versatile group of phloem-limited plant pathogens. '*Ca.* Liberibacter' spp. are insect-transmitted, and some are associated with severe diseases, but others cause only mild to asymptomatic infections in several plant hosts [1]. '*Ca.* L. asiaticus' (CLas) causes Huanglongbing (HLB) or citrus "greening" disease, which is arguably the single most devastating disease of citrus worldwide. HLB is also associated with '*Ca.* L. americanus' (CLam) in Brazil and '*Ca.* L. africanus' (CLaf) in Africa. Aberrant assimilate partitioning and nutrient transport in HLB-infected citrus trees leads to progressive decline in productivity and eventual death. '*Ca.* L. solanacearum' (CLso) causes zebra chip (ZC) disease on potato and is vectored by the tomato and potato psyllid *Bactericera cockerelli* Šulc (Triozidae) in America and New Zealand [2]. ZC symptoms include purple discoloration and chlorosis of leaves, stolon collapse, browning and necrotic flecking of vascular tissue in the tubers leading to rapid death of infected potato plants. CLso has a wide host range and can also infect several other economically important *Solanaceae* and *Apiaceae* crops, including tomato, carrot, fennel, celery, parsnip and parsley [3].

CLas is acquired by Asian citrus psyllid (ACP) *Diaphorina citri* Kuwayama (Psyllidae) nymphs while feeding and invades the midgut cells. The bacteria recruit endoplasmic reticulum-derived vacuolar vesicles as a unique replicative niche causing localized apoptosis in the midgut of ACP adults [4, 5]. The bacteria form an extensive biofilm on the outer midgut surface before planktonic release to the hemolymph *en route* to salivary gland invasion. Ultrastructural examination of CLso-infected *B. cockerelli* suggests colonization of insect midgut cells via an endo/exocytosis-like mechanism [6]. Both CLas and CLso spread inter- and intracellularly in psyllids across multiple organs at relatively high titers but without manifesting overt pathogenic symptoms [5, 6]. By contrast in plants, systemic colonization by CLas [7, 8] and CLso [9] is limited to phloem.

Active bacterial motility is essential for recognition and colonization of host tissues, nutrient resourcing and pathogenesis by many symbiotic and pathogenic bacteria [10, 11]. The complex helical or rotary nanomechanical action of flagella enables directed movement of bacterial cells through aqueous environments either by unicellular swimming or multicellular swarming [12]. Twitching motility is mediated by extension, tethering and retraction of certain Type IV pili (T4P) resulting in forward pulling of bacterial cells under conditions of low water content e.g., within biofilms [13]. Coordinated mechanosensing of host surfaces by rotary flagella and retractable T4P is essential for permanent bacterial attachment [14] and transitioning between biofilm-restricted sessile and planktonic phases [15]. T4P are also involved in biofilm architecture [16], adsorption and uptake of extracellular dsDNA for food and for horizontal gene transfer by naturally competent bacteria [13, 17, 18].

T4P are traditionally categorized into three subclasses, T4aP, T4bP and T4cP, based on morphological, serological, functional and biochemical criteria [19]. T4aP are widespread across several Gram-negative pathogenic and environmental bacteria such as *Pseudomonas*, *Neisseria*, *Myxococcus* etc. T4bP are restricted to the enteropathogenic (EPEC) and enterotoxigenic (ETEC) strains of *Escherichia coli*, *Vibrio cholerae* and *Salmonella enterica* serovar Typhi. T4aP contain NMePhe as the first N-terminus amino acid (aa) residue of the pilin subunit, a characteristic short (5–6 aa) leader peptide and 150–160 aa long mature length. By contrast, T4b pilins lack a conserved N-terminus NMePhe, have longer (15–30 aa) leader peptides and result in 180–200 aa mature pilin subunits [19, 20].

The third subclass, T4cP, includes Tad (Tight adherence) and Flp (Fimbrial low-molecular-weight-protein pili). The Tad pilus apparatus (homologous to *Caulobacter crescentus* pilus

assembly or Cpa) has been implicated in surface anchoring, adhesion, colonization, cellular aggregation and biofilm cohesion [17, 19, 21, 22]. Notably, Tad pili are not considered to be dynamic because they all lack a dedicated retraction ATPase for catalyzing pilin depolymerization, a prerequisite for generating the directional retractile force sufficiently large for active bacterial motility and uptake of extracellular dsDNA in naturally competent bacteria [23]. However, the CpaF motor ATPase of *C. crescentus* was demonstrated to drive repetitive bidirectional cycles of Tad pilus extension and retraction via ATP hydrolysis, resulting in "walking-like" surface translocation of bacteria in microfluidic chambers [14, 24].

Only one Liberibacter species, *L. crescens* (Lcr), derived from a diseased mountain papaya, has been axenically cultured *in vitro* [25], and has no known plant or insect host. Lcr strain BT-1 (GenBank Acc. NC_019907.1) is naturally competent for transformation and is well established as a culturable surrogate for the pathogenic '*Ca*. Liberibacter' spp., none of which have been axenically cultured [26]. Lcr BT-1 and all 35 sequenced genomes representing all '*Ca*. Liberibacter' spp. have been annotated as encoding a "nearly" complete set of biosynthetic genes for flagella; all are missing only three minor flagellins: FlaB, FlaC and FlaD [27, 28]. Similarly, a complete set of Tad pilus genes is present and conserved in all Liberibacters [29]. Differential regulation of the CLas Tad pilus operon was described and a potential role for the Tad pilus in bacterial attachment to the ACP midgut was suggested [29]. However, no other potential functional role for these pili in any member of the genus has been suggested. Evidence for surface appendages and their role(s) in motility and systemic colonization in '*Ca*. Liberibacter' spp. remains largely anecdotal.

In this report we provide ultrastructural evidence for the presence of both flagella and Tad pilus surface appendages in Lcr and demonstrate both flagellar swimming and Tad pilus-mediated twitching motility in the genus *Liberibacter* for the first time. Based on the presence, transcriptional activity and functional characterization of genes encoding flagella and Tad pili in Lcr, we propose that motility is an essential attribute of pathogenic '*Ca*. Liberibacter' spp. that likely determines circulative propagation in the psyllid as well as systemic colonization of phloem. More importantly, the demonstration of twitching motility in this genus extends the importance of Tad pili in '*Ca*. Liberibacter' spp. beyond their traditional role in tenacious surface adherence. Based on (a) the conservation of Tad pilus and DNA uptake genes, (b) our earlier observation of DNA uptake and natural competence in Lcr [26] and (c) the fact that all Liberibacters are deficient in nucleotide biosynthesis [30], we propose that Tad pilus-mediated uptake of extracellular dsDNA for food is likely essential for survival and colonization of '*Ca*. Liberibacter' spp. in both plant and insect hosts.

## Materials and methods

### Bacterial strains and growth conditions

The relevant characteristics, source and/or reference for the bacterial strains and plasmids used in this study are listed in Table 1. *E. coli* was grown in Luria-Bertani (LB) medium at 37˚C. Lcr BT-1 was maintained on BM7A medium containing 20 g l$^{-1}$ N-(2-acetamido)-2-aminoethanesulfonic acid (ACES) buffer (Sigma-Aldrich, St. Louis, MO, USA) [26] with gentle shaking at 110 rpm at 28˚C. Antibiotics were used as needed at the following concentrations (in µg/ml): 100 ampicillin (Amp) and 50 kanamycin (Kn) for *E. coli*; 4.5 Kn, 2.0 chloramphenicol (Cm) and 2.0 gentamycin (Gm) for Lcr.

### Transmission electron microscopy (TEM)

Copper grids (400 mesh, 62 µm pitch; Sigma-Aldrich) were placed with the Formvar side down, for 10 sec on a 7-day-old culture of Lcr growing on BM7A medium containing 0.25%

**Table 1. Bacterial strains and plasmids used in this study.**

| Strains[a] / plasmids | Relevant characteristics[b] | Reference/ source |
|---|---|---|
| Strains | | |
| *E. coli* TOP10 | F–*mcr*A Δ(*mrr-hsd*RMS-*mcr*BC) Φ80*lac*ZΔM15 Δ*lac*X74 *rec*A1 *ara*D139 Δ(*ara leu*) 7697 *gal*U *gal*K *rps*L (Str[R]) *end*A1 *nup*G | Invitrogen |
| Lcr BT-1 | Wild-type strain isolated from Babaco mountain papaya | [25] |
| Lcr FlgF-1 | *flgF*::pCLL026 derived from BT-1, Kn[R] | This study |
| Lcr FlgF-2 | *flgF*::pCLL026/pCLL038 derived from BT-1, Kn[R], Gm[R] | This study |
| Lcr FlgK-1 | *flgK*::pCLL027 derived from BT-1, Kn[R] | This study |
| Lcr FlgK-2 | *flgK*::pCLL027/pCLL036 derived from BT-1, Kn[R], Gm[R] | This study |
| Lcr CpaF-1 | *cpaF*::pCLL043 derived from BT-1, Kn[R] | This study |
| Lcr CpaA-1 | *cpaA*::pMJ081 derived from BT-1, Kn[R] | This study |
| Lcr CpaB-1 | *cpaB*::pMJ082 derived from BT-1, Kn[R] | This study |
| Lcr TadC-1 | *tadC*::pMJ083 derived from BT-1, Kn[R] | This study |
| Lcr ComEC-1 | *comEC*::pMJ054 derived from BT-1, Kn[R] | This study |
| Lcr ComEC-2 | *comEC*::pMJ054/pCLL052 derived from BT-1, Kn[R] | This study |
| Lcr ComEC-3 | *comEC*::pMJ054/pCLL053 derived from BT-1, Kn[R] | This study |
| Lcr ComEC-4 | *comEC*::pMJ054/pCLL056 derived from BT-1, Kn[R] | This study |
| Plasmids | | |
| pCR®2.1-TOPO | 3.9 kb; PCR cloning vector, Ap[R], Kn[R] | Invitrogen |
| pBBR1MCS-5 | Rep *Bordatella*, Mob+, *lacZ*, Gm[R] | [32] |
| pUFR071 | RepW, ColE1, Mob+, lacZ, Par+, derivative of pUFR040 lacking *Eco*RI site, Cm[R], Gm[R] | [62] |
| pUFZ075 | P$_{tryp}$-TIR-GFP cassette in pUFR034, Kn[R] | [31] |
| pCLL02 | 2 kb internal fragment of Lcr restriction subunit R *RIP* (B488_RS03405) in pCR®2.1-TOPO, Ap[R], Kn[R] | [26] |
| pCLL04 | Gm acetyl transferase gene *aacC1* (834 bp) inserted within the Lcr *RIP* fragment in pCLL02, Ap[R], Kn[R], Gm[R] | [26] |
| pCLL026 | 491 bp fragment of Lcr flagellar hook-associated protein gene *flgF* (B488_RS00930) in pCR®2.1-TOPO, Ap[R], Kn[R] | This study |
| pCLL027 | 707 bp fragment of Lcr basal-body rod protein gene *flgK* (B488_RS04530) in pCR®2.1-TOPO, Ap[R], Kn[R] | This study |
| pCLL035 | P$_{tryp}$ from pUFZ075 cloned downstream of *lacZ* in pBBR1MCS-5, Gm[R] | This study |
| pCLL036 | Full-length (1449 bp) Lcr *flgK* in pCLL035, Gm[R] | This study |
| pCLL038 | Full-length (729 bp) Lcr *flgF* in pCLL035, Gm[R] | This study |
| pCLL043 | 600 bp fragment of Lcr Tad pilus motor ATPase gene *cpaF* (B488_RS06240) in pCR®2.1-TOPO, Ap[R], Kn[R] | This study |
| pCLL044 | 769 bp internal fragment of Lcr Tad pilus biogenesis ATPase gene *cpaE* (B488_RS06245) in pCR®2.1-TOPO, Ap[R], Kn[R] | This study |
| pCLL052 | Full-length (2349 bp) Lcr inner membrane competence channel protein gene *comEC* (B488_RS05330) with native promoter in pCLL035, Gm[R] | This study |
| pCLL053 | Full-length (1185 bp) "*Ca.* L. asiaticus" psy62 *comEC* (CLIBASIA_RS01170) with native promoter in pCLL035, Gm[R] | This study |
| pCLL056 | Full-length (1114 bp) "*Ca.* L. solanacearum" ISR100 *comEC* (C0030_002320) with native promoter in pCLL035, Gm[R] | This study |
| pMJ054 | 701 bp fragment of Lcr *comEC* in pCR®2.1-TOPO, Ap[R], Kn[R] | This study |
| pMJ081 | 477 bp fragment of Lcr prepilin peptidase gene *cpaA* (B488_RS06265) in pCR®2.1-TOPO, Ap[R], Kn[R] | This study |
| pMJ082 | 417 bp fragment of Lcr Tad pilus periplasmic subunit gene *cpaB* (B488_RS06260) in pCR®2.1-TOPO, Ap[R], Kn[R] | This study |
| pMJ083 | 425 bp fragment of Lcr Tad pilus stabilization protein gene *tadC* (B488_RS06230) in pCR®2.1-TOPO, Ap[R], Kn[R] | This study |

[a] Lcr, *L. crescens* BT-1 (GenBank Acc. NC_019907.1).

[b] Ap, ampicillin; Cm, chloramphenicol; Gm, gentamycin; Kn, kanamycin; Str, streptomycin.

or 0.75% agar (w/v; Difco™ Agar, BD diagnostics, VWR, Radnor, PA, USA). Grids were carefully lifted off and the adherent bacterial cells were fixed/stained in a drop of 2% (v/v) uranyl acetate for 60 sec. Excess uranyl acetate was then blotted off and the residue allowed to air dry. Grids were viewed at 100 kV accelerating voltage using a transmission electron microscope (Hitachi H-7000, Hitachi High-Technologies Corporation, Tokyo, Japan) equipped with a Veleta (2k×2k) CCD side mount camera.

## Quantification of motility in *L. crescens* BT-1

Motility assays were performed in BM7A medium solidified with either 0.25% or 0.75% agar for quantification of swimming or twitching phenotype, respectively [12]. Five μl of 5-day-old Lcr culture grown in BM7A medium ($OD_{600}$ = 0.4) were spot-inoculated on the surface of agar plates. Likewise, 5 μl bacterial inoculum was also stabbed either halfway through the agar, or to the bottom of the petri plate for quantification of twitching phenotype. The inoculated plates were cultured for 3 weeks at 28°C and examined for bacterial motility away from the inoculated site [12]. In case of asymmetric growth, the colony diameter was measured at the widest point. Due to the severely reduced growth of the *cpaF* mutant strains, the primary transformation colonies were resuspended in liquid BM7A medium ($OD_{600}$ = 0.4) and used for motility assays. All experiments were repeated at least three times with three replications. Student's *t* tests (*P* = 0.05) were performed to separate treatment means.

## Site-directed mutagenesis and complementation of insertional mutants in *L. crescens* BT-1

Lcr flagellar genes (*flgF* and *flgK*), Tad pilus genes (*cpaA*, *cpaB*, *cpaE*, *cpaF*, *tadC*) and competence protein gene *comEC* were interrupted with a kanamycin-resistance gene and the resulting mutant strains were verified by PCR analyses following the strategy outlined in Jain et al. [26]. Partial DNA fragments internal to the coding regions of target genes were PCR-amplified using primer pairs described in Table 2 and the amplicons were cloned into non-replicative (in Lcr) plasmid pCR®2.1-TOPO (Invitrogen, Carlsbad, CA, USA). The resulting suicide plasmids (Table 1) were transformed into Lcr by electroporation.

The $P_{tryp}$ promoter from pUFZ75 (Kn[R]) [31] was amplified using the primers CLL35F/R and inserted downstream of the *lacZ* promoter in pBBR1MCS-5 [32] within the *Kpn*I/*Xho*I sites resulting in plasmid pCLL035 (Gm[R]). Full-length *flgF* and *flgK* were PCR-amplified from Lcr using primer pairs CLL38F/R and CLL36F/R, respectively. After sequence verification, the *flgF and flgK* inserts were directionally subcloned using *Bam*HI/*Xba*I into pCLL035 and the resulting plasmids pCLL038 and pCLL036 were transformed into mutant Lcr strains *flgF*::pCLL026 and *flgK*::pCLL027, respectively, for complementation. Full-length *comEC* was PCR-amplified using the primer pairs CLL52F/R, CLL53F/R and CLL56F/R from Lcr, CLas strain psy62 [33] and CLso strain ISR100 [34], respectively. Sequence-verified inserts were directionally cloned into pCLL035 using *Xma*I/*Bam*HI sites and the resulting plasmids (pCLL052, pCLL053 and pCLL056) were transformed into Lcr strain *comEC*::pMJ054.

## Natural transformation of *L. crescens* BT-1

A 40 μl aliquot of electrocompetent Lcr BT-1 cells [26] was mixed gently with 0.5 μg closed circular plasmid DNA (pUFR071) and 50 μl chitin (1 mM, C-3387, Sigma-Aldrich) or xanthan gum (0.5% w/v in water; G-1253, Sigma-Aldrich). The transformation mix was resuspended in 850 μl liquid BM7A medium, incubated with gentle shaking at 28ºC at 150 rpm for 48 hrs and finally plated on selective BM7A medium containing 4.5 mg l[-1] Kn and 2.0 mg l[-1] Gm. Natural uptake of pUFR71 by *comEC*::pMJ054 strains carrying the complementing full-length *comEC* was selected in the presence of 4.5 mg l[-1] Kn and 2.0 mg l[-1] each of Gm and Cm.

## Nucleic acid extraction and quantitative reverse transcription PCR (qRT-PCR) analyses

For Lcr cultures grown on agar plates, the bacteria were gently scraped off the plates and resuspended in 100–200 μl water and briefly centrifuged to collect the cells in the pellet. Genomic

**Table 2. Primers used in this study.**

| [a]Target / primer | Forward primer sequence (5'→3')[b] | Reverse primer sequence (5'→3')[b] |
|---|---|---|
| Cloning in pCR®2.1-TOPO for marker interruption | | |
| Lcr *flgF*_internal | CGTTAGAGCGTCGATTGACTAC | CCTGCAATGGTTCTTCTTTAGG |
| Lcr *flgK*_internal | GCACTACTAAAGAGCCCTGAAA | AGGATGCTTATGGGAGGAAATG |
| Lcr *cpaA*_internal | GCTGTCCGGATAGGATATAAAACCG | GTGATTCCTGCTATCGTTTTTCTTGT |
| Lcr *cpaB*_internal | CTCCTTGGAATCAGCTGTAGAA | GATGGCGCTGTAATACGTTTG |
| Lcr *cpaE*_internal | AAATCATTCGGTCGCTCTTT | GAATACCAGACTTTGGTTTC |
| Lcr *cpaF*_internal | CGGGTTGATGAATCTAGTCC | AGGTAGACTAAATCCTCCCA |
| Lcr *tadC*_internal | CGTTCAGTCCGACCTTCCTCAG | GCACAAAAGCCTCTAATGGTTCGTG |
| Lcr *comEC*_internal | TCCGGTTCTACCTCGTCTTT | TGAGAAGAGGGTGAGAGATAGC |
| Cloning in pBBR1MCS-5 for complementation | | |
| P*tryp* (CLL35F/R) | *GGTACC*GCTCTAAGAAGCTTGGCAAA | *CTCGAG*AGGCATGCAAGCTTGGCACC |
| Lcr *flgK* (CLL36F/R) | *TCTAGA*TCAGCTGGAAAAAATCTTGA | *GGATCC*ATGTCTCTTTCAGCGGCTTT |
| Lcr *flgF* (CLL38F/R) | *TCTAGA*TCAGCTGGAAAAAATCTTGA | *GGATCC*ATGTCTCTTTCAGCGGCTTT |
| Lcr *comEC* (CLL52F/R) | *CCCGGG*TAATATTTCA ATTATATAAC | *GGATCC*TTAGTCTTTGAATTGTGGTT |
| CLas *comEC* (CLL53F/R) | *CCCGGG*TCAAATAGCGTGAACCCTTA | *GGATCC*TTATATTCTATGCTTGGTCC |
| CLso *comEC* (CLL56F/R) | *CCCGGG*CAGTAACTACTTATTTTTTA | *GGATCC*TCAAAATAAACTGTCTATTC |
| Quantitative reverse-transcription polymerase chain reaction | | |
| Lcr *flhA* | GAAATCCCTCACGCCGTATATC | GGTTATCGTTGGTTCAGGAGAG |
| Lcr *flgC* | CGATACTGATGATGGGAAGTTACA | CCAGCAGCTATCGTGCTAAT |
| Lcr *flgE* | GTACCATCTACTGGCTCCTATTC | CCTTCACCATCAATCGCTAAATC |
| Lcr *flgF* | AATAATGTGGGCACCAAGATTTC | CGCAAACCAGACATTACCTTTC |
| Lcr *flgH* | GAGATTCTCAAGCTGCTTTGTTT | TACGACTCCGGCCAGTATTA |
| Lcr *flgK* | GTAACAGGATTCTGACGGACAT | CTTTCAGCGGCTTTGAATAAGG |
| Lcr *cpaA* | CAGAAACGCTGAGACCTAAGAA | TTCGGATGGGATTCATCTCTTG |
| Lcr *cpaB* | AGAATTGGATGGCGCTGTAATA | CGCTTTCCAGAGGGAAGAATAG |
| Lcr *cpaC* | GGCTCAGCTAGAGTACGAATTAC | GGAAGTAGCGGTCAACAAGAA |
| Lcr *cpaE* | GGCACAACGAACAAGCAAA | AGGATCCAGCACAAGGAATATC |
| Lcr *cpaF* | AAACTTCTGGGCCTCTAACTTC | AAACGCGGCCTCCAAATA |
| Lcr *tadC* | CTGCAAGAGGAGCAGATTGT | TGGCCTGATGCTCTTGATTT |
| Lcr *prfA* | AGGCTCAAGTTGGTTCAGG | ATATCACCCTCCAACATGCG |
| Lcr *gyrB* | TCTTCACCAGCATCTCCAAC | CACTTTCATCTTGGCTGCG |

[a]Lcr, *L. crescens* BT-1 (GenBank Acc. NC_019907.1); CLas, 'Ca. L. asiaticus' psy62 (GenBank Acc. NC012985.3); CLso, 'Ca. L. solanacearum' ISR100 (GenBank Acc. NZ_PKRU02000006.1). Flagellar biosynthesis protein *flhA* (B488_RS04500), basal-body rod protein *flgC* (B488_RS00990), hook protein *flgE* (B488_09460), basal-body rod protein *flgF* (B488_RS00930), basal body L-ring protein *flgH* (B488_RS01020) and hook-associated protein *flgK* (B488_RS04530); Tad pilus prepilin peptidase *cpaA* (B488_RS06265), periplasmic subunit *cpaB* (B488_RS06260), secretin *cpaC* (B488_RS06255), biogenesis ATPase *cpaE* (B488_RS06245), motor ATPase *cpaF* (B488_RS06240) and stabilization protein *tadC* (B488_RS06230); inner membrane competence channel protein *comEC* (Lcr, B488_RS05330; CLas, CLIBASIA_RS01170; CLso, C0030_002320); DNA gyrase subunit B *gyrB* (B488_RS06700) and peptide chain release factor-1 *prfA* (B488_RS00360).
[b]Restriction sites for cloning are underlined and *italicized*.

DNA and RNA were extracted using the GenElute™ Bacterial Genomic DNA kit (Sigma-Aldrich) and RNeasy® Mini Kit (Qiagen, Valencia, CA, USA) following manufacturer's recommendations. RNA was diluted with nuclease free water to 200 ng µl⁻¹ and cleaned with TURBO DNA-free (DNase) Kit (Ambion, Austin, TX, USA). First-strand cDNA was synthesized from one µg purified RNA template using iScript Advanced cDNA Synthesis Kit (Bio-Rad, Hercules, CA, USA). Twenty µL of qPCR reaction consisted of 10 µl of 2× QuantiNova SYBR Green PCR Mix (Qiagen), 1 µl of each primer (5 µM), 4 µl of cDNA template and 5 µl of

DNase/RNase free water. The thermal cycling protocol entailed an initial activation step at 95˚C for 2 min, followed by 40 cycles of 95˚C for 5 s and 60˚C for 10 s. All cDNA samples were run in triplicates using a CFX96 Touch Real-Time PCR detection system (Bio-Rad). Optical data were acquired during the annealing step and the fidelity of PCR reaction was monitored by melting curve analysis beginning at 55˚C through 95˚C, ramped at $0.1˚C\ s^{-1}$. The gene-specific primers are listed in Table 2. Transcript levels were normalized using chromosomal reference gene DNA gyrase subunit B (*gyrB*, B488_RS06700) and transcript values in liquid BM7A were used as calibrator controls ($2^{-\Delta\Delta Ct}$). The data were analyzed using SAS for Windows, version 9.2 (SAS Institute Inc, Cary, NC) and Student's *t* tests were performed to separate treatment means. A probability of 5% was used to determine statistically significant differences.

## Fluorescence labeling and uptake of DNA

The *Bam*HI/*Sma*I digested linear insert (2.834 kb) from pCLL04 [26] carrying the internal fragment of the Lcr *RIP* gene (Type I restriction endonuclease subunit R; B488_RS03405) and Gm resistance gene (aminoglycoside-3-*O*-acetyltransferase-I, *aacC1*; 834 bp) from pUFR071 was used for fluorescence labeling of DNA and uptake via natural transformation. Dimeric cyanine nucleic acid dye YOYO™-1 iodide ($\lambda_{ex/em}$ 491/509; Invitrogen) was used for noncovalent fluorescent labelling of DNA at a base pair to dye ratio of 1:50. One µg $ml^{-1}$ YOYO-1-labeled DNA was pretreated with 10 units of DNase I (Sigma-Aldrich) for 10 min and added to the Lcr cell suspension in 100 µl BM7A ($OD_{600} \sim 0.4$) and incubated for 15–45 min at 28˚C. The fluorescence data were captured using an Olympus IX81-DSU Spinning Disc Confocal Microscope (Olympus Corporation, Tokyo, Japan) fitted with Hamamatsu ORCA-Flash 4.0LT + camera (Hamamatsu Photonics, K.K., Hamamatsu, Japan).

## Results

### Both flagella and much smaller Tad pili were observed by TEM of *L. crescens* BT-1

Both flagella and pili were readily distinguished upon ultrastructural examination of Lcr cells. Negatively stained Lcr cells grown on BM7A medium containing 0.25% (w/v) agar revealed the presence of peritrichous flagella distributed on the surface of rod-shaped ($\sim 0.5 \times 1.75$ µm), asymmetrically dividing bacterial cells. Pili were observed on the surfaces of Lcr cells grown on medium solidified with 0.75% (w/v) agar. By comparison with long, threadlike, electron dense flagella (Fig 1A and 1B), the pili were smaller, shorter and appeared relatively electron lucent (Fig 1C and 1D). Since the only pilus genes encoded by the Lcr genome are Tad pilus genes, we concluded these were Tad pili. Notably, among all the Lcr cell cultures examined, only a small subset of the cell population presented flagella or Tad pili on their surface, and these were observed only on cells grown on semisolid BM7A medium. Neither flagella nor Tad pili were observed on the surfaces of Lcr cells growing in a continuously shaking BM7A broth culture.

### Both swimming and twitching motility were observed with *L. crescens* BT-1

Low (0.25%) and medium (0.75%) agar (w/v) concentrations were optimized to differentiate between swimming and twitching motility [12]. Lcr grown in a low (0.25%) agar medium demonstrated swimming behavior as uniform outward growth from the inoculation point along both the surface (Fig 1E) and downward (Fig 1F) within the culture medium. By contrast, twitching motility of Lcr was evident at medium (0.75%) agar concentrations as surface

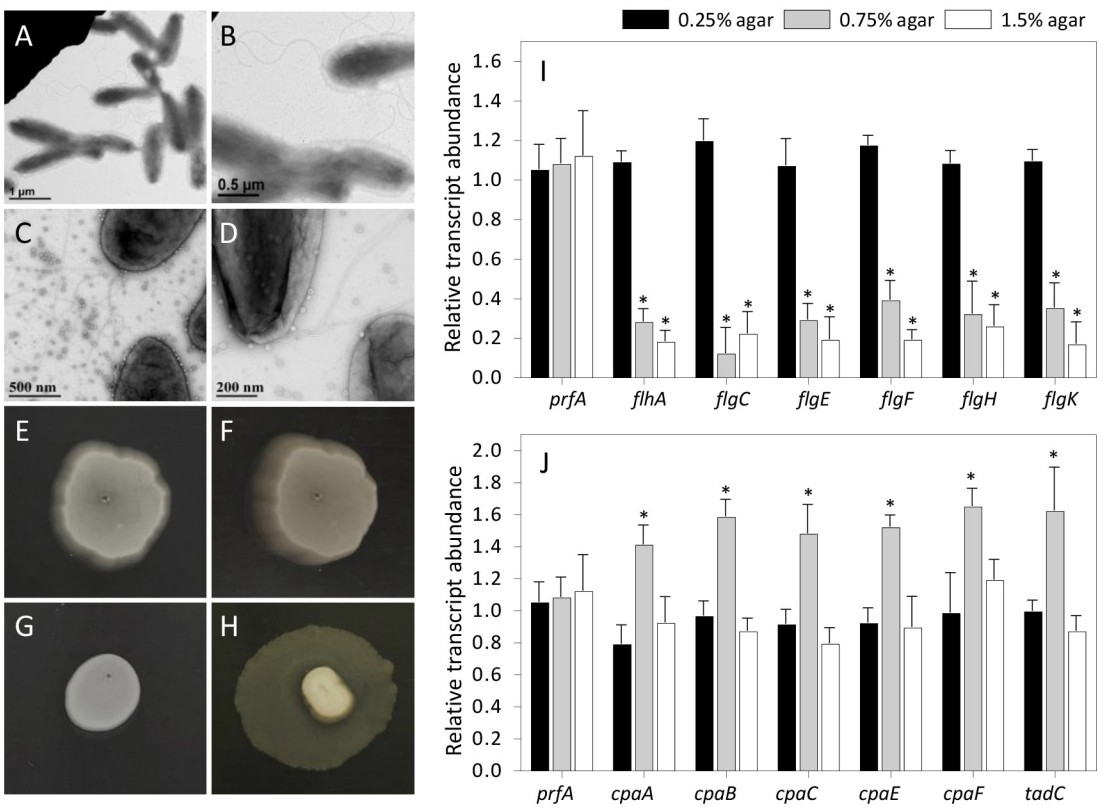

**Fig 1. Swimming and twitching motility in *L. crescens* strain BT-1.** (A-D) Ultrastructural and (E-H) functional evidence for the presence of flagella and Tad pili in Lcr BT-1. Five-day-old Lcr cells were cultured for three weeks on BM7A medium containing 0.25% (A-B, E-F) and 0.75% (C-D, G-H) agar for swimming and twitching assays, respectively. Negatively stained cells were analyzed by transmission electron microscopy. *Magnification* scales are indicated in the images. Note the swimming zone as outward bacterial growth along the surface (E) and clearly downward within the culture medium when the same colony is photographed at an angle (F). By contrast, Tad pilus-mediated twitching appeared as strictly surface restricted growth (G) or along the agar/polystyrene interface when the inoculum was stabbed to the bottom of the plate (H). Relative expression of (I) flagella and (J) Tad pilus biosynthesis genes in Lcr grown in BM7A broth culture and on BM7A plates containing low (0.25%), medium (0.75%) and high (1.5%) agar concentrations. The transcript abundance of each gene was calibrated against expression in liquid BM7A medium and normalized against the expression levels of the chromosomal reference gene *gyrB* within each sample. Bars represent average ± the standard deviation for three independent experiments with four replications. *Asterisks* represent significant differences ($P < 0.05$) in the transcript abundance.

restricted growth if the inoculum was surface-dropped or stabbed half-way through the medium (Fig 1G) and restricted along the medium/polystyrene interface if stabbed to the bottom of the plate (Fig 1H).

## Expression of flagella and Tad pilus genes in *L. crescens* BT-1 was associated with swimming and twitching, respectively

Expression of genes encoding flagella and Tad pili was examined in Lcr cultured in BM7A medium at three agar concentrations; low (0.25%, associated with swimming), medium (0.75%, associated with twitching) and high (1.5%, used for routine microbial maintenance and culturing). By comparison with Lcr grown in liquid cultures, qRT-PCR analyses demonstrated that the transcription of flagellar biosynthesis genes was significantly downregulated in Lcr cells when cultured on medium (0.75%) or high (1.5%) concentrations of agar but not on 0.25% agar (Fig 1I), associated with the swimming phenotype (Fig 1E and 1F). By contrast, the

transcript abundance of the Tad pilus apparatus genes was upregulated compared to liquid BM7A medium only at medium (0.75%) agar concentrations (Fig 1J), associated with the twitching phenotype seen under these conditions (Fig 1G and 1H). No significant changes were observed in the expression level of endogenous housekeeping control gene peptide chain release factor-1 (*prfA*, B488_RS00360) in cells growing under either liquid or semisolid culture conditions.

## Site-directed mutagenesis and complementation of flagella genes affected swimming but not twitching motility in *L. crescens* BT-1

Insertional mutagenesis of flagellar basal-body rod protein gene *flgF* (B488_RS00930) and hook-associated protein gene *flgK* (B488_RS04530) caused 20% and 33% reduction in the swim zone diameters of respective mutant cells *flgF*::pCLL026 (1.03 ± 0.08 cm) and *flgK*::pCLL027 (1.19 ± 0.07 cm) in comparison to the wild-type BT-1 (1.51 ± 0.19 cm). The full-length Lcr genes *flgF* and *flgK* fully complemented their respective mutant strains *flgF*::pCLL026/pCLL038 (1.33 ± 0.07 cm) and *flgK*::pCLL027/pCLL036 (1.48 ± 0.12 cm) (Fig 2A, 2B and 2E). However, no difference in the twitching zone diameters were observed for the wild-type BT-1 (1.03 ± 0.02 cm), *flgF*::pCLL026 (1.01 ± 0.04 cm), *flgF*::pCLL026/pCLL038 (1.01 ± 0.02 cm), *flgK*::pCLL027 (1.02 ± 0.02 cm) and *flgK*::pCLL027/pCLL036 (1.03 ± 0.01 cm) (Fig 2C–2E). Even though both swimming and twitching motility were observed with wild-type BT-1 (Fig 1), only swimming motility was affected by flagellar mutations (Fig 2). These data clearly show that Lcr was indeed capable of twitching motility independently of the flagella apparatus, and such translocation was deduced to be mediated by Tad pili. The functional role of Tad pilus genes in twitching motility and its association with ComEC-mediated natural competence in Lcr was therefore examined.

## Two distinct Tad pilus ATPase-encoding genes, *cpaE* and *cpaF*, are structurally and functionally conserved among all Liberibacters

The Tad pilus operon of all Liberibacters encodes two structurally and functionally distinct genes organized in tandem, a pilus biogenesis ATPase *cpaE* (B488_RS06245) and a motor ATPase *cpaF* (B488_RS06240) located immediately downstream. Comparative phylogenetic analyses showed that both CpaE (S1 Fig) and CpaF (Fig 3) are conserved across all α-Proteobacterial lineages examined, including all pathogenic '*Ca*. Liberibacter' spp. and cultured Lcr.

ClustalW [35] analysis revealed that Lcr CpaE (427 aa, 47.3 kDa) has only limited (53–60%) identity between CpaE homologs from '*Ca*. Liberibacter' spp. and only 42% with *C. crescentus* and *Rhizobium* spp. CpaE belongs to the SIMIBI [after signal recognition particle (SRP), MinD, and BioD] ATPase subclass of the large P-loop GTPase superfamily [36] and features an ATPase domain (aa position 164–426) with a deviant Walker A motif, KGGVGSS (aa position 173–179) (S1 Fig).

## Activity of all Liberibacter CpaF motor ATPases is likely bidirectional

The Tad pilus motor ATPase CpaF (481 aa, 53.4 kDa) is a soluble and cytoplasmic P-loop NTPase domain superfamily protein [37]. By contrast with CpaE, the CpaF homologs of Lcr, CLas, CLam, CLaf, CLso, *C. crescentus* and *Agrobacterium rhizogenes* are all highly similar (app. 70% identical; Fig 3).

ClustalW analysis showed the presence of multiple conserved domains, including an ATP-binding structural domain (aa positions 77–435) with conserved ATP-binding residues L[228] and D[403] and canonical nucleotide phosphate-binding G-rich Walker A (GGTGSGKT) and

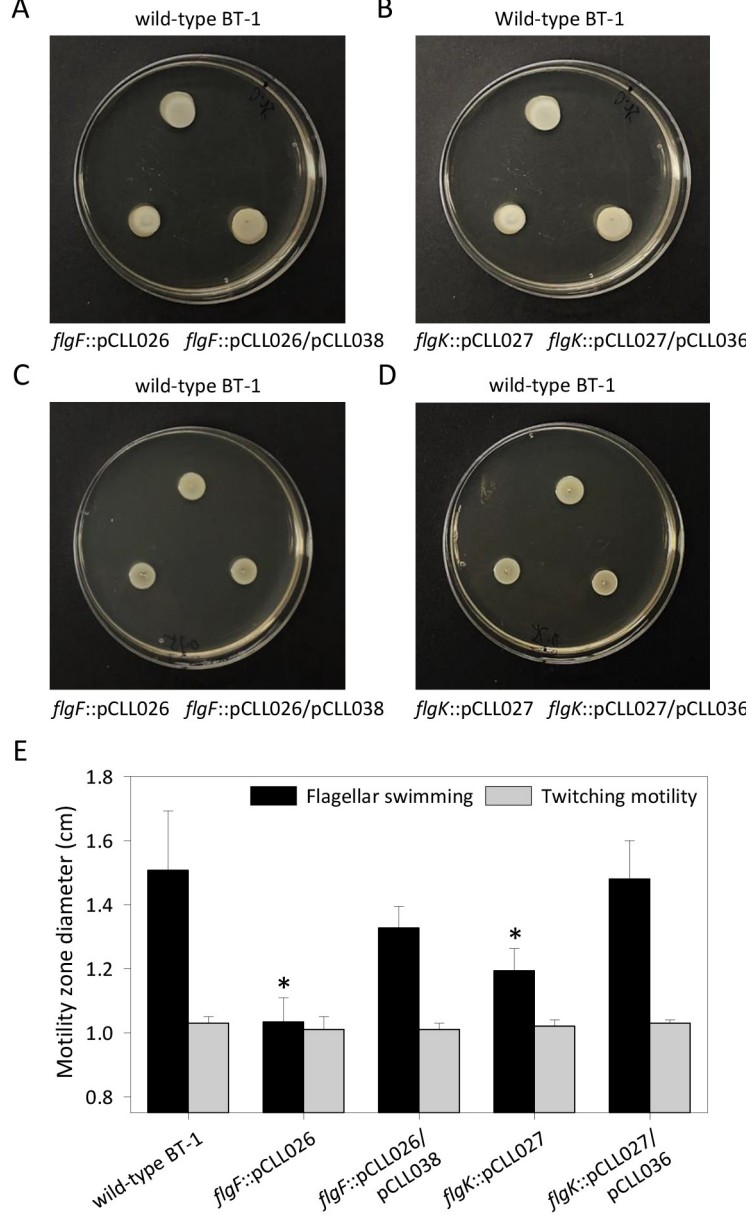

**Fig 2. Mutations in the *L. crescens* BT-1 flagellar genes *flgF* and *flgK* affected only swimming motility.** (A, B) Swimming and (C, D) twitching motility in the wild-type BT-1, insertional flagellar gene mutants (*flgF*::pCLL026 and *flgK*::pCLL027) and complemented (*flgF*::pCLL026/pCLL038 and *flgK*::pCLL027/pCLL036) strains on semisolid BM7A plates containing 0.25% and 0.75% agar for swimming and twitching assays, respectively. (E) Swimming and twitching motility zone diameters were measured in three independent assays with three replicates each. The data are average ± standard deviation and the significant differences ($P < 0.05$, Student's *t* test) in swimming behavior are represented by *asterisks*.

Walker B (RIILGE) motifs present between positions 256–263 and 327–332, respectively (Fig 3). Invariant residues $T^{263}$ of Walker A and $E^{332}$ of Walker B were present in Lcr CpaF and conserved among all Liberibacters; these residues are known to form coordinate $Mg^{2+}$ bonds between the β- and γ-phosphate moieties of ATP bound to the conserved $K^{262}$ of Walker A [38]. Three invariant acidic residues $E^{283}$, $D^{284}$ and $E^{287}$ (within the Asp Box motif) and the

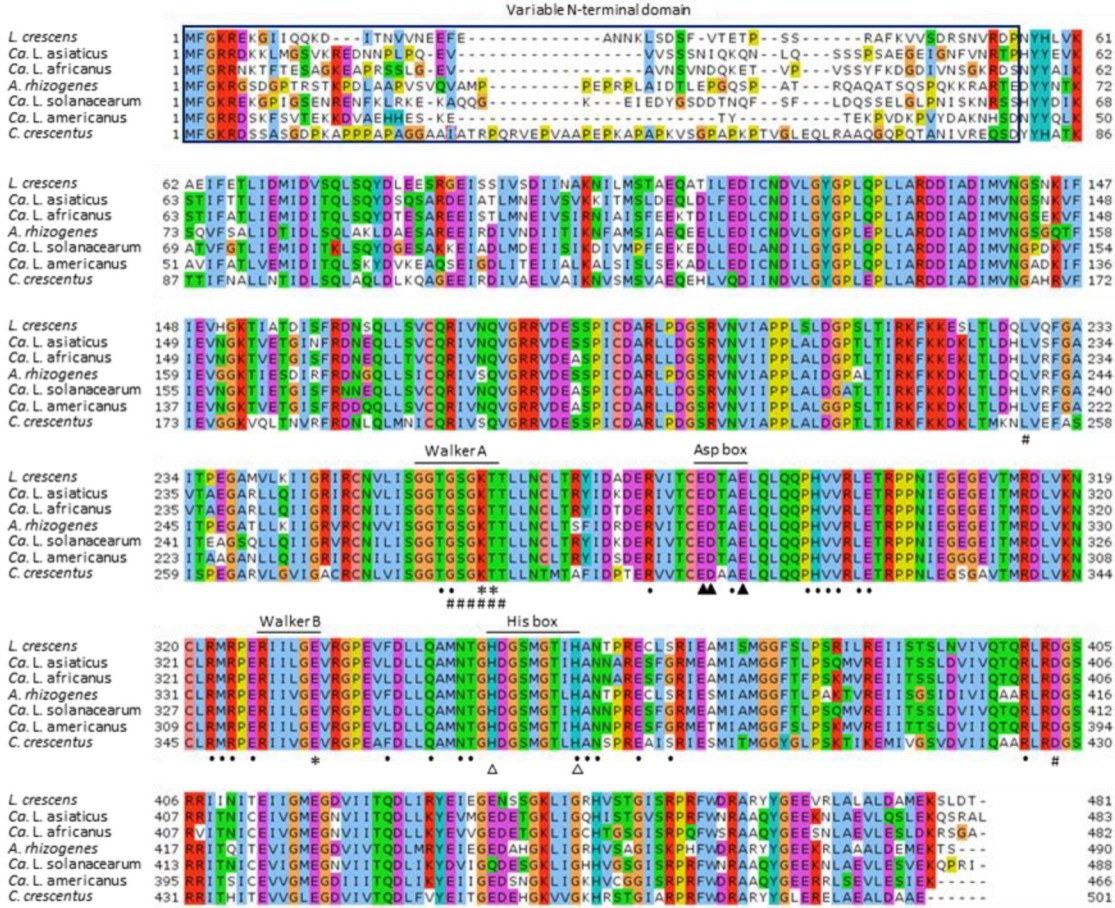

**Fig 3. Tad pilus motor ATPase CpaF is structurally conserved among all Liberibacters.** Sequence alignment of CpaF encoded by *L. crescens* BT-1 (WP_015273698.1), '*Ca.* L. asiaticus' (WP_015452438.1), '*Ca.* L. africanus' (WP_047264073.1), *A. rhizogenes* (WP_065114603.1), '*Ca.* L. solanacearum' (WP_076969128.1), '*Ca.* L. americanus' (AHA27602.1) and *C. crescentus* (YP_002518410.1). Invariant residues required for coordination of $Mg^{2+}$ between β- and γ-phosphate moieties of ATP within the canonical ATP-binding Walker A and Walker B motifs are marked by an asterisk (*). Invariant residues in Asp box and His box motifs are denoted by filled (▲) and open (△) triangles, respectively. Amino acids participating in ATP binding and ATPase activity are denoted by hashtags (#) and those involved in hexamer interface for subunit interactions are denoted by dots (·).

two invariant residues $H^{349}$ and $H^{357}$ (within the His Box motif) are considered essential for P-loop ATPase activity [23]. All Liberibacter CpaF homologs are therefore likely to form a toroidal homohexameric ring-like structure involved in pilin translocation across the inner membrane via "open" and "closed" conformational changes typical of known P-loop secretion NTPases [22]. Strict conservation of all functional domains (87% query coverage, 70.48% identity, E value 0.0; Fig 3) between Lcr and *C. crescentus* CpaF [14, 24] indicated that the activity of the Lcr CpaF motor ATPase and by extension, of all Liberibacters, is likely bidirectional and capable of generating the strong retractile force required for twitching motility.

## Mutations in Tad pilus genes affected twitching motility and abolished growth in *L. crescens* BT-1

To functionally validate the role of Tad pili in Lcr twitching motility, site-directed mutations were attempted in genes encoding the pilus biogenesis ATPase *cpaE*, motor ATPase *cpaF*,

prepilin peptidase *cpaA* (B488_RS06265), periplasmic subunit *cpaB* (B488_RS06260) and pilus stabilization protein *tadC* (B488_RS06230). Insertional mutants of *cpaA*, *cpaB*, *cpaF* and *tadC* were all severely compromised in growth and very few visible colonies were obtained, and then only after 3–4 months of culture on selective medium. Almost no colonies were subculturable, and even these colonies were not amenable to repetitive subcultures on either solid or liquid media. Despite several attempts, site-directed mutagenesis of *cpaE* failed to yield any viable colonies. It was also extremely difficult to achieve adequate density of viable cells of mutant strains of Tad pilus genes in liquid cultures to permit further experiments. Multiple attempts to prepare competent cells for complementation via electroporation failed. By contrast, mutations of non-essential Lcr genes typically yield viable colonies in 8–10 weeks [26].

Only *cpaF* insertional mutant strain (*cpaF*::pCLL043) was evaluated for twitching phenotype on BM7A plates containing 0.75% agar. A significant reduction of 26% in twitching zone diameter of *cpaF* mutant strain (1.04 ± 0.06 cm) was observed as compared to the wild-type BT-1 (1.43 ± 0.07 cm) (Fig 4B and 4C). On 0.25% agar plates, significantly altered swimming behavior was clearly observed deep into the agar by the *cpaF* mutant as compared to wild-type BT-1 (Fig 4A). However, when measured across the surface of the agar only, the *cpaF* mutant swimming zones (1.25 ± 0.03 cm) on 0.25% agar plates were comparable to those of wild-type BT-1 (1.26 ± 0.04 cm) (Fig 4A and 4C). Further, a significant difference in growth density was also observed between the mutant and wild-type BT-1 on 0.25% agar plates. Impairment of the twitching phenotype in the *cpaF* mutant strain of Lcr (Fig 4B and 4C), taken together with the high levels of protein sequence similarity among CpaF motor ATPases (Fig 3) confirmed that Tad pili in Lcr and likely in all Liberibacters, are dynamic and involved in surface translocation.

## ComEC, necessary for natural competence, is not conserved among '*Ca. Liberibacter*' spp.

Demonstrably dynamic Tad pili in Lcr and the severity of the growth defects observed with mutations of Lcr Tad pilus genes suggested a correlation between Tad pili and both natural transformation in Lcr [26] and DNA uptake for food in all Liberibacters [30], since dsDNA uptake in bacteria strictly requires a transport assembly through the outer membrane to the periplasm [20].

The adsorption and periplasmic uptake of extracellular dsDNA through the outer membrane is mediated through retraction of T4P, followed by translocation of long ssDNA into the cytoplasm via a conserved *comEC/rec2*-encoded inner membrane channel protein [39, 40] (also refer Fig 6). Bioinformatic analysis of Lcr *comEC* (B488_RS05330) (S2 Fig) predicted that ComEC (782 aa, 87.5 kDa) has an N-terminal domain of unknown function *DUF4131* (Pfam accession PF13567) followed by a universal transmembrane *competence* domain (Pfam accession PF03772), an architecture commonly seen in $\alpha$-Proteobacteria [40]. ComEC in Lcr and '*Ca. Liberibacter*' spp. lacked the metallo-β-lactamase (*Lactamase_B*) domain (Pfam accession PF00753) responsible for the degradation of a single strand in the dsDNA that is frequently found in a broad family of DNA and RNA nucleases [39]. The *DUF4131* domain (aa position 87–241) participates in oligonucleotide-binding (OB fold) and the *competence* domain (aa position 291–579) includes a set of core transmembrane helices mediating the transport of ssDNA across the inner membrane (S2 Fig). Topology modeling (using TMHMM-2.0) revealed that the mature ComEC was predicted to be an integral inner membrane protein with 11 transmembrane helices (aa position 61–83, 88–107, 111–133, 311–333, 353–375, 396–418, 454–476, 489–511, 526–545) and a conserved divalent metal-binding motif, GLSHIIAISGLN

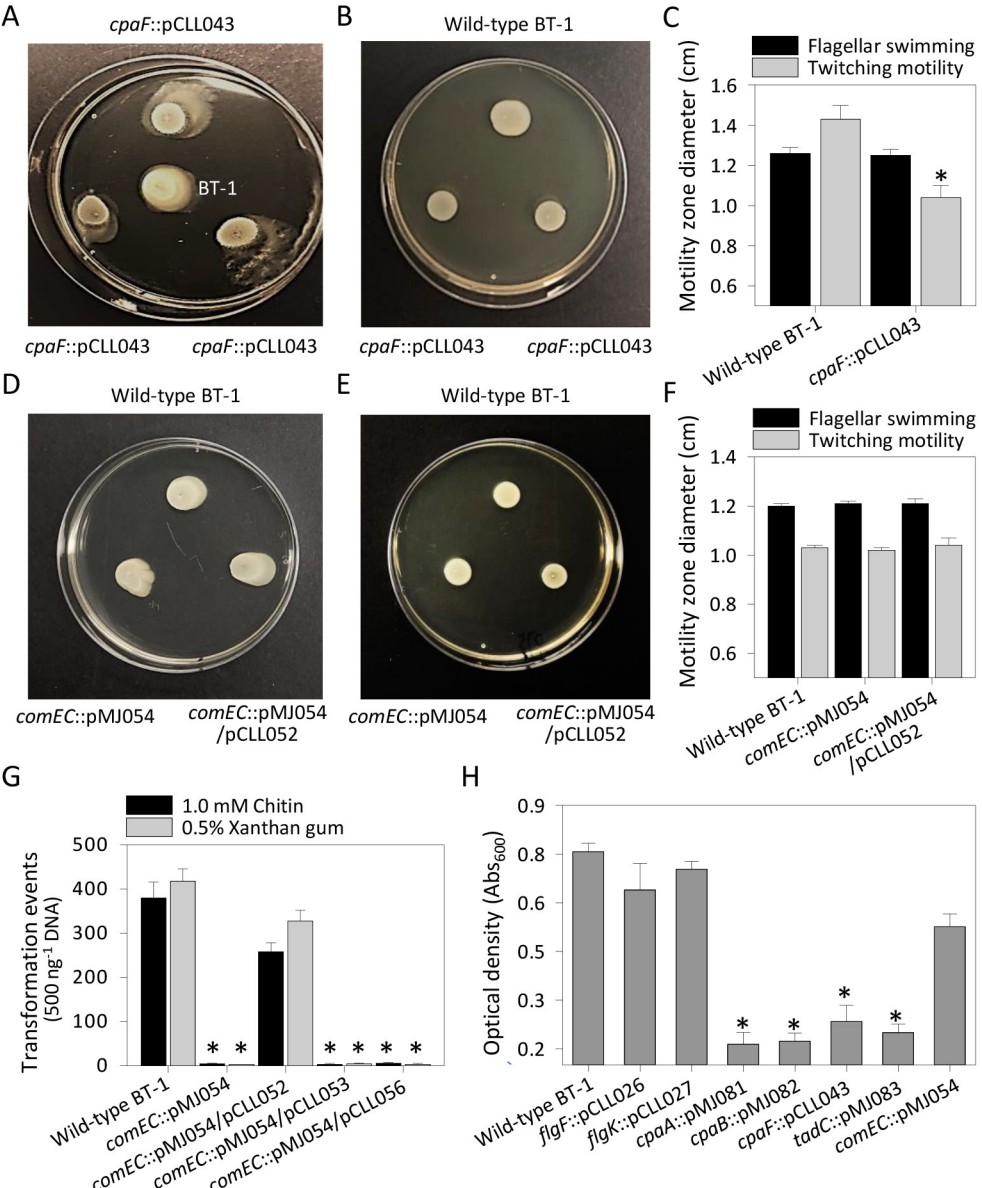

**Fig 4. Mutations in Tad pilus genes affected twitching motility and growth but *comEC* mutation affected only natural transformation in *L. crescens* BT-1.** (A-C) Mutation in the Tad pilus motor ATPase gene *cpaF* had no effect on flagellar swimming (A, C) but reduced twitching motility (B, C) in wild-type BT-1 and the mutant *cpaF*::pCLL043. (D-F) Mutation in the inner membrane competence protein gene *comEC* had no effect on flagellar swimming (D, F) or twitching motility (E, F) but completely abolished natural competence for transformation (G). Deficiency in natural competence for transformation in the mutant strain *comEC*::pMJ054 was fully complemented by full-length *comEC* gene from Lcr (*comEC*::pMJ054/pCLL052) but not from CLas strain psy62 (*comEC*::pMJ054/pCLL053) or CLso strain RS100 (*comEC*::pMJ054/pCLL056) (G). (H) Insertional mutant strains of Tad pilus genes (*cpaA*::pMJ081, *cpaB*::pMJ082, *cpaF*::pCLL043 and *tadC*::pMJ083) were severely compromised in growth in culture as compared to wild-type BT-1, and insertional mutants for flagellar genes (*flgF*::pCLL026 and *flgK*::pCLL027) and the competence protein gene (*comEC*::pMJ054). Motility zone diameters and natural competence for transformation were quantified in three independent experiments with three replicates each. Bacterial growth assays were repeated thrice with five replicates each. The data are average ± standard deviation and the significant differences ($P < 0.05$, Student's *t* test) in bacterial motility, natural transformation and growth are represented by *asterisks*.

(aa position 311–322, containing an invariant H[314]) [41], embedded within the fourth transmembrane helix.

Phylogenetic analyses revealed that only truncated or cryptic homologs of *comEC* are found in most, if not all, '*Ca*. Liberibacter' spp. Potentially full-length ComEC homologs were found only in the CLso strains RSTM [42] and LsoNZ1 [43], CLaf strain PTSAPSY [44] and CLam strain São Paulo [45], all containing the oligonucleotide-binding *DUF4131* and the universally conserved transmembrane *competence* domains (S2 Fig).

## Mutations in *comEC* affected natural transformation in *L. crescens* BT-1, but not motility or growth

A site-directed insertion in *comEC* had no effect on either swimming (1.21 ± 0.01 cm) (Fig 4D and 4F) or twitching (1.02 ± 0.01 cm) (Fig 4E and 4F) phenotypes as compared to wild-type BT-1 (1.20 ± 0.02 cm and 1.03 ± 0.01 cm, respectively). Wild-type BT-1 had an average natural transformation efficiency of about 380 ± 36 (using 1 mM chitin) and 418 ± 28 (using 0.05% xanthan gum) antibiotic resistant colonies per 500 ng of closed circular pUFR071 plasmid DNA (Fig 4G). However, mutation of *comEC* nearly abolished natural competence for transformation in the mutant strain (*comEC*::pMJ054), yielding only 2–3 transformation events under identical conditions. The full-length Lcr *comEC* gene rescued the natural competence phenotype in the mutant strain *comEC*::pMJ054/pCLL052 yielding 258 ± 20 (using 1 mM chitin) and 328 ± 24 (using 0.05% xanthan gum) antibiotic resistant colonies. Loci annotated as *comEC* from CLas strain psy62 [33] and CLso strain ISR100 [34] were found to be nonfunctional, yielding only 1–5 naturally transformed colonies from either pCLL053 (*comEC* from CLas strain psy62) or pCLL056 (*comEC* from CLso strain ISR100) in several independent experiments.

While mutations of Tad pilus genes *cpaA*, *cpaB*, *cpaF* and *tadC* resulted in mutant strains that were severely compromised in growth as compared to wild-type BT-1, mutation of flagellar genes (*flgF* and *flgK*) and *comEC* had no effect on growth in culture (Fig 4H). Addition of sheared salmon sperm DNA (10 µg ml$^{-1}$ BM7A medium) did not alleviate the growth deficiency of the Tad pilus mutant strains.

## The *L. crescens* BT-1 Tad pilus mediates uptake of extracellular dsDNA

To visualize DNA entry into Lcr cells, a 2.8 kb linear dsDNA fragment was fluorescently labelled with the noncovalent dye YOYO-1. YOYO-1 remains virtually non-fluorescent in aqueous solutions while specifically staining dsDNA by bis-intercalation at base pair to dye ratios of 1:8. After 15 min exposure to fluorescently labeled dsDNA in the presence of DNase I, 48.4 ± 4.2% of the wild-type BT-1 (Fig 5A) and 44.6 ± 4.0% of *comEC*::pMJ054 (Fig 5I) imported dsDNA into the cells. In contrast, *cpaF*::pCLL043 cells, lacking retractile Tad pili, had only transiently visible fluorescent foci (15.3 ± 5.9% and 5.9 ± 4.9% cells following DNase I treatment for 15 and 30 min, respectively) (Fig 5E and 5G). While the DNA fluorescence remained stable in wild-type BT-1 (Fig 5D) and *comEC*::pMJ054 (Fig 5L) cells, fluorescent foci were virtually absent in *cpaF*::pCLL043 cells after 45 min of DNase I treatment (Fig 5H), due to digestion of DNA outside of the cells.

A working model illustrating the Liberibacter Tad pilus structure, function and conserved gene organization is presented in Fig 6. The data presented herein summarize the unique and critical role of the bifunctional motor ATPase CpaF and several other Tad pilus components in twitching motility and the associated periplasmic uptake and ComEC-mediated cytoplasmic translocation of environmental dsDNA in Lcr for food and natural transformation.

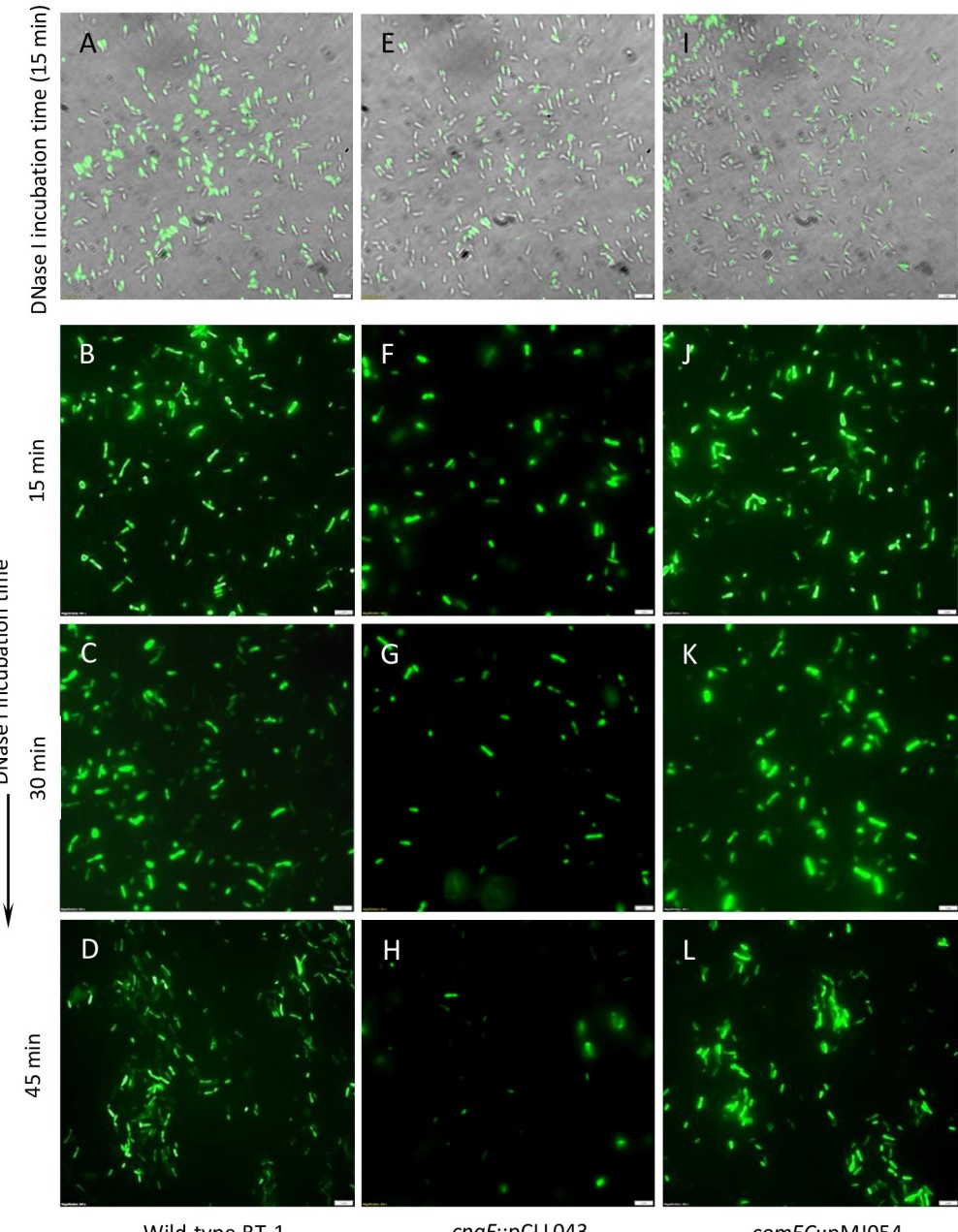

**Fig 5. Tad pilus-mediated uptake of fluorescently labeled dsDNA by *L. crescens* wild-type BT-1, *cpaF* and *comEC* insertional mutant strains.** (A, E and I) Overlay of bright-field and YOYO-1 florescence in Lcr wild-type BT-1, *cpaF* and *comEC* insertional mutant strains (*cpaF*::pCLL043 and *comEC*::pMJ054, respectively) exposed to YOYO-1-stained dsDNA (1 μg ml⁻¹) for 15 min. Uptake of YOYO-1-stained fluorescent dsDNA by wild-type BT-1, *cpaF*::pCLL043 and *comEC*::pMJ054 cells after DNase I exposure for 15 min (B, F and J), 30 min (C, G and K) and 45 min (D, H and L), respectively. Fluorescent DNA foci were transiently observed in *cpaF*::pCLL043 cells whereas the DNA fluorescence was stable (resistant to DNase I) in wild-type BT-1 and *comEC*::pMJ054 cells. YOYO-1-stained dsDNA was also pretreated with DNase I for 10 min prior to starting the experiment. *Magnification* scale = 2 μm.

## Discussion

Active motility and dispersal are fundamental processes that allow bacteria to intercept, explore and colonize host tissue, evade host immune responses, reduce competition for

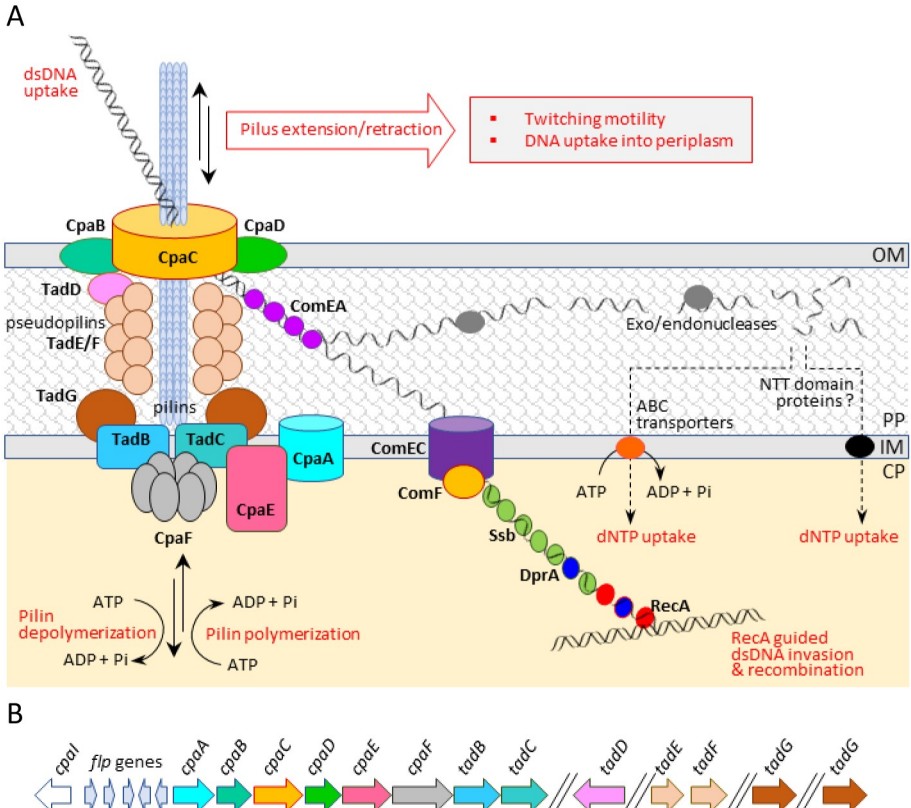

**Fig 6. Tad pilus-mediated twitching motility is functionally coupled with periplasmic uptake of extracellular dsDNA.** (A) A working model for Tad pilus-mediated uptake of extracellular dsDNA into periplasm and ComEC-mediated cytoplasmic translocation of ssDNA in naturally competent Lcr [26]. Bidirectional motor ATPase CpaF drives pilus extension and retraction via pilin polymerization and depolymerization causing both the twitching motility as well as DNA uptake. Schematic illustration of Tad pilus has been modified from Denise et al. [20]. (B) The structural components of the Tad pilus are named following Andrade and Wang [29]. Essential Tad pilus genes in Lcr encode the biogenesis ATPase (*tadZ*/*cpaE*, B488_RS06245), motor ATPase (*tadA*/*cpaF*, B488_RS06240), prepilin peptidase (*tadV*/*cpaA*, B488_RS06265), inner membrane staging complex (*tadB*/*cpaG*, B488_RS06235 and *tadC*/*cpaH*, B488_RS06230), secretin (*rcpA*/*cpaC*, B488_RS06255), periplasmatic subunits (*tadG*/*cpaB*, B488_RS06260 and *rcpB*/*cpaD*, B488_RS06250), inner membrane anchor (*rcpC*/*tadG*, B488_RS05235) and pilotin (*tadD*/*cpaO*, B488_RS05560). The genes encoding the DNA translocation machinery include dsDNA receptor (*comEA*, B488_RS00175), inner membrane channel protein (*comEC*, B488_RS05330), ATP-dependent translocase (*comF*, B488_RS05095), ssDNA binding proteins (*ssb*, B488_RS01885), DNA processing protein A (*dprA*, B488_RS04595) and the recombinase protein (*recA*, B488_RS02195). While Lcr is naturally competent for transformation, in this model, all Liberibacters are presumed to likely assimilate extracellular dsDNA as a food source. Except for *comEC*, rest of the Tad pilus and natural competence components are well conserved between Lcr and '*Ca*. Liberibacter' spp. Canonical NTT domain containing nucleotide transporter proteins remain to be identified in Lcr and/or '*Ca*. Liberibacter' spp. Abbreviations: OM, outer membrane; IM, inner membrane; PP, periplasm; CP, cytoplasm.

nutrient resourcing and avoid adverse niche environments. Despite significantly reduced (~1.2 Mb) genomes, all 35 sequenced strains of CLas [33], CLam [45] and CLso [2] contain a 'minimal but complete' and functional [27, 29, 46, 47] set of flagellar and Tad pilus biosynthetic genes. The present report provides the first empirical evidence for active flagellar swimming and Tad pilus-mediated twitching motility in Lcr, a phylogenetically related and an established surrogate host for functional genomic studies of the uncultured pathogenic '*Ca*. Liberibacter' spp.

## *Ca*. Liberibacter sp. flagellar genes are suppressed *in planta*

Innumerable ultrastructural studies of HLB pathogens within their plant hosts have failed to provide any evidence for either flagella or pili to date. Highly coordinated and regulated expression of nearly 30–100 flagellar proteins, their assembly and the rotary function imposes a steep energy drain on the bacterial cell [48]. As opposed to planktonic growth in the psyllid hemolymph, where flagella may be required, CLas and CLso colonization is strictly intracellular *in planta* where energy intensive flagellar swimming is likely not needed. Pathogenic '*Ca*. Liberibacter' spp. are incapable of uptake and utilization of sucrose and scavenge ATP from host cell cytoplasm [49] and may therefore, selectively downregulate flagellar biosynthesis to avoid the associated energy drain. It is also conceivable that bacterial transmission into plant phloem cells triggers shedding of flagella, and flagellin expression is repressed as a strategy to prevent PAMP (Pathogen-associated Molecular Pattern) -associated host innate immune responses [29, 50–53].

## Tad pilus-mediated twitching motility is required in both psyllid and plant hosts

Chemo- and mechanosensing of tissue surfaces, adhesive forces and the rheology of its surroundings by bacteria is facilitated through Tad pilus-mediated twitching and aggregation [15]. CLas and CLso Tad pilus genes are differentially upregulated in psyllids as compared to the plant hosts [29, 47, 54] and indicate the functional importance of Tad pilus-mediated adherence and motility in insect endosymbiosis. Replicative and circulative colonization by CLas and CLso in insect hosts requires Tad pilus mediated spatial recognition and penetration of psyllid midgut tissue, biofilm formation on the hemocoel surface of midgut, planktonic bacterial growth in the hemolymph, and finally recognition and penetration into the salivary glands' lumen [4, 6]. Colonization of psyllid salivary glands reaching above a threshold bacterial titer is a prerequisite for efficient transmission of CLas [4] and CLso [55]. Tad pilus-mediated twitching may also explain uneven long-distance cell-to-cell movement of CLas through the plasmodesmata pore units (sieve pore/plasmodesmata complex). The observed uneven spatial and temporal distribution of CLas in citrus [56] argues against strictly passive bacterial movement via bulk symplastic solute flow within the plant hosts. Adherence of CLas to the plasma membrane of the phloem cells specifically adjacent to the sieve pores [7] lends support in favor of Tad-pilus mediated translocation across sieve pore units.

## Tad pili in *Ca*. Liberibacter sp. are retractile and required for twitching and DNA uptake

The current dogma limits the function of Tad pili exclusively to surface adherence, and only the T4aP participate in motility and DNA uptake. Dynamic movement of T4aP in the bacterial outer membrane and periplasm is orchestrated by dedicated cytoplasmic extension (PilB) and retraction (PilT) ATPases belonging to the Additional Strand Catalytic 'E' (ASCE) subfamily of AAA+ ATPases [17, 23]. While PilT is dispensable for piliation [23], both PilB and PilT are required for twitching motility and natural competence [57]. PilB and PilT have no known structural or functional homologs encoded within the highly conserved Tad pilus operons of Lcr and '*Ca*. Liberibacter' spp. Ellison et al. [24] provided the seminal evidence for retractile and dynamic *C. crescentus* Tad pili driven by the bidirectional activity of CpaF motor ATPase and suggested acquisition of *cpaF* genes across several bacterial species via widespread horizontal gene transfer events. Phylogenetic and structural similarity between Lcr and *C. crescentus* motor ATPase CpaF proteins (Fig 3) is consistent with Tad pili being involved in active

twitching motility (Fig 4B and 4C) as well as in DNA uptake (Fig 5E–5H). These data extend the function of Tad pili beyond surface adherence in Lcr and pathogenic 'Ca. Liberibacter' spp.

A universal prerequisite for all natural competence systems is a structural supramolecular assemblage, such as T4a pili or Type 2 (or rarely Type 4) Secretion Systems (T2SS, T4SS) for transport of DNA across the outer membrane into periplasm and an integral DNA translocation complex for cytoplasmic transport [20]. Natural competence for transformation via uptake of both linear and circular plasmid DNA has previously been demonstrated in Lcr [26]. Adsorption, uptake and cytoplasmic translocation of extracellular dsDNA can readily be explained by a functional and retractile Tad pilus, given the absence of any other T4P or competence pili in any Liberibacter genome. To the best of our knowledge, a requirement for *tad*-like genes for natural transformation has earlier been shown only in the case of *Micrococcus luteus* [58], even though the molecular mechanism of retraction was not examined.

## Tad pilus-mediated DNA uptake is essential for viability of all Liberibacters

Despite multiple attempts, we were unable to mutagenize the pilus biogenesis ATPase gene *cpaE*. Pilus biogenesis protein CpaE is required for efficient localization of the motor ATPase CpaF and is the first step of pilus assembly following an inside-out path [36]. Insertional mutation in the Tad pilus motor ATPase gene *cpaF* resulted in loss of twitching phenotype as well as severely reduced growth in culture (Fig 4B, 4C and 4H). Insertional mutants of *cpaA*, *cpaB*, *cpaF* and *tadC* were all severely compromised in growth (Fig 4H). Taken together, these data indicate that the dynamics of Tad pili and twitching phenotype are invariably associated with bacterial growth and viability. Although the bidirectional ATPase motor activity of *C. crescentus* CpaF was demonstrated to drive Tad pilus retraction required for surface sensing, orientation and attachment during biofilm formation [14, 24], a role of CpaF activity in uptake of extracellular dsDNA was not investigated. Both piliation and the CpaF-dependent adsorption and acquisition of extracellular dsDNA by Tad pili (Fig 5) may be crucial for Lcr growth since the enzyme systems for the metabolism of purines and pyrimidines are lacking in the highly reduced Liberibacter genomes [30]. The acquired extracellular dsDNA can be repurposed as the sole C and N source supporting microbial growth [59] or the constituent nucleotides can be salvaged for nucleic acid biosynthesis [60].

## Tad pilus mediated DNA uptake is primarily for food in Liberibacters

The *cpaF* mutation in Lcr severely affected twitching motility (Fig 4B and 4C) and bacterial growth but had no effect on surface-restricted swimming (Fig 4A and 4C). Nonetheless, these poorly growing *cpaF* mutant cells (Fig 4A and 4H) exhibited an unusually enhanced swimming motility deep into 0.25% agar as compared to the wild-type BT-1 (Fig 4A) likely triggered by nutrient seeking behavior. Flagellar swimming activity is well documented to be stimulated by nutritional need [12]. Several obligate intracellular parasites with reduced genomes encode proton-energized nucleotide transporter (NTT) proteins that mediate net import of nucleotides for DNA replication and energy parasitism compensating for the loss of ATP generation and nucleotide biosynthesis [61]. Except for the *nttA*-encoded ATP/ADP translocase present only in 'Ca. Liberibacter' spp. [49], additional NTT domain containing proteins remain to be identified in Liberibacters. Comparative overview of the COG functional groups revealed 12 additional genes allocated to nucleotide transport and metabolism in CLso haplotype D as compared to Lcr [34].

## ComEC mediates DNA uptake for natural competence in *L. crescens* BT-1

The *comEC* mutation in Lcr had no effect on bacterial motility functions (Fig 4D–4F) or growth (Fig 4H) but completely abolished natural competence for transformation (Fig 4G). Integral ComEC is required for mediating periplasmic DNA binding, single strand degradation to expose the other strand and passage of ssDNA across the membrane resulting in horizontal gene transfer [39] (refer Fig 6). Since only truncated or cryptic homologs of *comEC* are found in most, if not all, strains of '*Ca*. Liberibacter' spp. sequenced to date (S2 Fig and Fig 4G), it is likely that natural competence is limited to only a few strains of Liberibacters, including Lcr.

## Conclusions

Active bacterial motility is critical for several aspects of '*Ca*. Liberibacter' pathogenesis, including acquisition and transmission by the insect vector as well as systemic colonization of both the insect and plant hosts. Fig 6 summarizes the dynamic activity of Tad pilus motor ATPase CpaF, a prerequisite both for motility as well as DNA uptake. We propose that the active process of Tad pilus-mediated uptake of extracellular dsDNA likely fulfills the nucleotide requirements for '*Ca*. Liberibacter' spp. highlighting the need for externally supplied DNA and a source of ATP [49]. Nucleotide enrichment of culture media may therefore be required to overcome one of many metabolic bottlenecks for successful axenic culturing of pathogenic '*Ca*. Liberibacter' spp. Conversely, chemical inhibition of the biosynthesis of the sole pilus in '*Ca*. Liberibacter' spp. would adversely impact pathogen motility and nucleotide acquisition, making this pilus and its components prime targets for disease control.

## Supporting information

**S1 Fig. Essential motifs of Tad pilus biogenesis ATPase CpaE are conserved among all Liberibacters.** Sequence alignment of CpaE encoded by *L. crescens* BT-1 (WP_015273699.1), '*Ca*. L. asiaticus' (WP_015452437.1), '*Ca*. L. africanus' (WP_047264074.1), '*Ca*. L. solanacearum' (WP_103846917.1), '*Ca*. L. americanus' (WP_144079396.1), *Rhizobium* spp. CRIBSB (WP_166603675.1) and *Caulobacter crescentus* (YP_002518411.1). Deviant ATP-binding Walker A motif is marked. Amino acids participating in ATP binding and ATPase activity are denoted by hashtag (#) and residues involved in dimer interface are denoted by dots (·). The 'signature' $K^{173}$ residue in Lcr CpaE is expected to mediate homodimerization by binding to the phosphates of ATP engaged by the other subunit. The deviant Walker A motif of all pathogenic '*Ca*. Liberibacter' spp. diverges further, where the 'signature' K residue is replaced by similar, positively charged and small sized R or H residues.
(TIF)

**S2 Fig. Alignment of ComEC sequences from *L. crescens* (Lcr) and '*Ca*. Liberibacter' spp.** The following sequences were used for alignment: Lcr strain BT1 (WP_015273513.1), '*Ca*. L. asiaticus' (Las) strains psy62 (WP_012778576.1), Ishi-1 (WP_045490146.1) and gxpsy (WP_012778576.1-like), '*Ca*. L. solanacearum' (Lso) strains ZC1 (WP_080550987.1), RI (ONI59307.1), FIN111 (WP_076969511.1), RSTM (WP_055347849.1) and NZ1 (KJZ82350.1), '*Ca*. L. africanus' (Laf) strain PTSAPSY (WP_052775004.1) and '*Ca*. L. americanus' (Lam) strain Sao Paulo (AHA27693.1). The N-terminal domain of unknown function *DUF4131* followed by the universal transmembrane *competence* domain are represented by black and red boxes, respectively. The metal-binding motif is underlined. The invariant $H^{314}$ and the signal peptide cleavage sites (in Lcr) are denoted by open (△) and filled (▲) triangles, respectively.
(TIF)

## Acknowledgments

We thank Patricia Rayside for excellent technical assistance and Dr. Ofir Bahar, Agricultural Research Organization, Volcani Center, Israel for providing gDNA sample of CLso strain ISR100.

## Author Contributions

**Conceptualization:** Mukesh Jain, Kathryn M. Jones, Michelle Heck, Dean W. Gabriel.

**Data curation:** Mukesh Jain.

**Formal analysis:** Mukesh Jain.

**Funding acquisition:** Kathryn M. Jones, Dean W. Gabriel.

**Investigation:** Lulu Cai, Mukesh Jain, Marta Sena-Vélez, Laura A. Fleites.

**Methodology:** Lulu Cai, Mukesh Jain, Marta Sena-Vélez, Kathryn M. Jones, Laura A. Fleites, Dean W. Gabriel.

**Project administration:** Dean W. Gabriel.

**Resources:** Dean W. Gabriel.

**Supervision:** Mukesh Jain, Dean W. Gabriel.

**Validation:** Marta Sena-Vélez, Kathryn M. Jones.

**Writing – original draft:** Mukesh Jain, Dean W. Gabriel.

**Writing – review & editing:** Lulu Cai, Mukesh Jain, Marta Sena-Vélez, Kathryn M. Jones, Laura A. Fleites, Michelle Heck, Dean W. Gabriel.

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
