## [Decision Letter · Decision Letter 0]

2 Sep 2021

PONE-D-21-23751

Tad pilus-mediated twitching motility is essential for DNA uptake and survival of Liberibacters.

PLOS ONE

Dear Dr. Gabriel,

Thank you for submitting your manuscript to PLOS ONE. After careful consideration, we feel that it has merit but does not fully meet PLOS ONE’s publication criteria as it currently stands. Therefore, we invite you to submit a revised version of the manuscript that addresses the points raised during the review process.

The MS is quite lengthy and needs shortening to improve the readability. Kindly refer to referee's suggestions and address carefully. Discussion and Introduction need to be shortened, and figures 5-6 should go to supplementals. Proposed Model must move to back of results to maintain the flow.

We look forward to receiving your revised manuscript.

Kind regards,

Tushar Kanti Dutta, Ph.D.

Academic Editor

PLOS ONE

Journal Requirements:

5. Please upload a new copy of Figure 4, 5 and 6 as the detail is not clear. Please follow the link for more information: https://blogs.plos.org/plos/2019/06/looking-good-tips-for-creating-your-plos-figures-graphics/

Additional Editor Comments:

The MS is lengthy and needs shortening to improve the readability. Discussion and Introduction need to be shortened, and figures 5-6 should go to supplementals. Model can be moved to back of results to maintain the flow.

Reviewers' comments:

Reviewer's Responses to Questions

**Comments to the Author**

1. Is the manuscript technically sound, and do the data support the conclusions?

Reviewer #1: Yes

2. Has the statistical analysis been performed appropriately and rigorously? 

Reviewer #1: Yes

3. Have the authors made all data underlying the findings in their manuscript fully available?

Reviewer #1: Yes

4. Is the manuscript presented in an intelligible fashion and written in standard English?

Reviewer #1: Yes

5. Review Comments to the Author

Reviewer #1: The manuscript ‘Tad pilus-mediated twitching motility is essential for DNA uptake and survival of Liberibacters’ by Cai et al. describes that the Liberibacter Tad pili are essential for twitching motility and DNA uptake for dsDNA in Liberibacter crescens, which is used as a surrogate for unculturable plant pathogenic Liberibacters.

Liberibacter are important plant pathogens. The manuscript is exhaustive, and authors have done a good job by providing the ultrastructural details of both- flagella as well as Tad pilus, and demonstrated the tad-pilus mediated uptake of extracellular DNA in L. crescens. This is important as Liberibacters cannot synthesise nucleotides and depend on DNA uptake for their survival.

The manuscript is based on a robust hypothesis, and the experimental evidence provided is adequate and sufficient to demonstrate that flagella and Tad pilus are present, are involved in swimming and twitching motility, respectively, and that the Tad pilus is involved in DNA uptake. The results from the mutation and complementation experiments supports the conclusions. The bioinformatic analysis is adequate and supports the conclusions.

I have following suggestion to improve the manuscript.

Introduction needs to be shortened.

The results, especially the model and the bioinformatics results (lines 310 to 335, lines 347-355) have become more of discussion than results. I suggest that the model should be moved to the end of results, and all the discussion presented in the bioinformatics results should be moved to discussion section. The model (Figure 3) should be presented at the end of results after the bioinformatic and mutational analysis to preserve the flow of the manuscript.

Figures 5 and 6 may be moved to supplemental information.

The discussion may be shortened a bit and organised into subheadings for better readability.

6. PLOS authors have the option to publish the peer review history of their article (what does this mean?). If published, this will include your full peer review and any attached files.

Reviewer #1: **Yes: **Vishal S. Somvanshi

---

## [Editor Report · Decision Letter 1]

1 Oct 2021

Tad pilus-mediated twitching motility is essential for DNA uptake and survival of Liberibacters.

PONE-D-21-23751R1

Dear Dr. Gabriel,

We’re pleased to inform you that your manuscript has been judged scientifically suitable for publication and will be formally accepted for publication once it meets all outstanding technical requirements.

Kind regards,

Tushar Kanti Dutta, Ph.D.

Academic Editor

PLOS ONE
---

## [Editor Report · Acceptance letter]

5 Oct 2021

PONE-D-21-23751R1 

Tad pilus-mediated twitching motility is essential for DNA uptake and survival of Liberibacters. 

Dear Dr. Gabriel:

I'm pleased to inform you that your manuscript has been deemed suitable for publication in PLOS ONE. Congratulations! Your manuscript is now with our production department. 

Kind regards, 

on behalf of

Dr. Tushar Kanti Dutta 

Academic Editor

PLOS ONE